# Generation of Multiple Vector Optical Bottle Beams

Svetlana N. Khonina [1,2,*], Alexey P. Porfirev [1], Sergey G. Volotovskiy [1], Andrey V. Ustinov [1], Sergey A. Fomchenkov [1,2], Vladimir S. Pavelyev [1,2], Siegmund Schröter [3] and Michael Duparré [3]

1. Image Processing Systems Institute—Branch of the Federal Scientific Research Centre, "Crystallography and Photonics" of Russian Academy of Sciences, 443001 Samara, Russia; porfirev.alexey@ipsiras.ru (A.P.P.); sv@ipsiras.ru (S.G.V.); andr@ipsiras.ru (A.V.U.); s.a.fom@mail.ru (S.A.F.); pavelyev10@mail.ru (V.S.P.)
2. Samara National Research University, 443086 Samara, Russia
3. Leibniz Institute of Photonic Technology (IPHT), 07745 Jena, Germany; siggi_schroeter@yahoo.de (S.S.); michael.duparre@leibniz-ipht.de (M.D.)
* Correspondence: khonina@ipsiras.ru

**Abstract:** We propose binary diffractive optical elements, combining several axicons of different types (axis-symmetrical and spiral), for the generation of a 3D intensity distribution in the form of multiple vector optical 'bottle' beams, which can be tailored by a change in the polarization state of the illumination radiation. The spatial dynamics of the obtained intensity distribution with different polarization states (circular and cylindrical of various orders) were investigated in paraxial mode numerically and experimentally. The designed binary axicons were manufactured using the e-beam lithography technique. The proposed combinations of optical elements can be used for the generation of vector optical traps in the field of laser trapping and manipulation, as well as for performing the spatial transformation of the polarization state of laser radiation, which is crucial in the field of laser-matter interaction for the generation of special morphologies of laser-induced periodic surface structures.

**Keywords:** cylindrical vector beams; vector optical bottle; diffractive optical elements; axicon; polarization

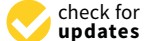



## 1. Introduction

In many applications, control over the polarization structure of the generated light field allows one to finely adjust light-matter interaction. For example, cylindrical vector beams (CVBs), a class of axially symmetric laser beams with spatially variant polarization [1], can improve the axial trapping efficiency of high-refractive-index particles by reducing the scattering force [2]. In addition, CVBs are superior for trapping particles bigger than the wavelength when the numerical aperture of the used microobjectives is higher than 1.1 [3]. The first-order CVBs, azimuthally and radially polarized laser beams, provide a tool for the realization of pulling/pushing laser beams for spherical light-absorbing particles [4]. In addition, the effect of the control of the local morphology of the formed laser-induced periodic surface structures (LIPSS) using vector beams with non-uniform polarization is well known [5,6]. Because of this, there is a need to develop efficient and scalable techniques for the generation of single and multiple vector laser beams with the desired polarization structure. The approaches for generation of single CVBs are well studied, and include the use of subwavelength gratings [7], metasurfaces [8], anisotropic crystals [9], different forms of the so-called *q*-plate [10] and *S*-waveplate [11], as well as dynamic spatial light modulators (SLMs) [12,13]. For splitting single CVBs into multiple identical beams, one-dimensional diffraction gratings are well proven [14]. It should be noted that the appropriate arrangement of multiple polarization singularities inherent to CVBs allows one to control the polarization distributions of the focal fields while the intensity maintains unchanged [15].

Diffractive optical elements (DOEs) are, in fact, a powerful tool for the generation of structured laser beams and transformation of CVBs, as previously and repeatedly

shown [16,17]. The use of DOEs makes it possible to obviate the challenges arising from the use of SLMs, namely their relatively low damage threshold and the need to use an additional optimization encoding for the realization of polarization changes [18,19]. In addition, the possibility of combining transmission functions of different DOEs in a single diffractive element allows one to design elements that generate a laser beam with a complex three-dimensional structure of intensity, phase, and polarization [20]. Nowadays, such structured three-dimensional light fields are actively used in laser material processing and laser manipulation [21,22]. The spatial-temporal characteristics of such structured laser beams are actively studied, too [20,23]. It should be noted that the dynamic control of the intensity distributions formed by the DOEs can be made by changing the polarization state of the illuminating radiation, which also significantly expands the field of application of diffractive optics [24–27].

Recently, we demonstrated and investigated the three-dimensional spatial intensity and polarization transformations in the focal region (at the sharp focusing and paraxial regimes) with the help of a combination of a lens and a diffraction axicon with different (linear, binary, and spiral) structures [28]. Such diffractive axicons are well-known elements for the generation of annular light distributions in the focal plane [29,30]. A binary version of the linear axicon can be used for the generation of a subwavelength longitudinally polarized beam with a large uniform depth of focus when the element is illuminated by a radially polarized laser beam [31]. In this paper, we consider the possibility of using DOEs for splitting a single vector beam and generating multiple polarization singularities, forming a three-dimensional light distribution of a vector "bottle" beam, a laser beam with an intensity null that is surrounded by an optical barrier in three dimensions [32]. For this purpose, different versions of diffractive linear and spiral axicons were encoded into single DOEs and fabricated using e-beam lithography. The theoretical and experimental studies of the arising spatial-polarization transformations are presented. The cases of different mutual locations of the generated annular and spiral-shaped vector laser beams with the desired polarization structure are investigated. With an increase in the distance from the focal plane, the diameters of the formed light rings and spirals decrease while the distances between the generated polarization singularity points stay the same. This should allow them to be used not only for laser trapping and stable three-dimensional guiding of nano- and microparticles, but also, for example, in applications of the controllable formation of arrays of LIPSS with tunable morphologies defined by the locations of the generated polarization singularity points.

## 2. Methods

### 2.1. Theoretical Analysis

For the investigation of the electric vector field in the focal region of an optical system consisting of a combination of a lens and a diffractive axicon, the simplified (in the paraxial regime) version of the Richards–Wolf equation [33–35] can be used in the following form:

$$
\mathbf{E}_\perp(\rho, \nu, z) = \begin{pmatrix} E_x(\rho, \nu, z) \\ E_y(\rho, \nu, z) \end{pmatrix} =
$$
$$
-\frac{if}{\lambda} \int\limits_0^{\theta_{max}} \int\limits_0^{2\pi} B(\theta, \varphi) \begin{pmatrix} c_x(\varphi) \\ c_y(\varphi) \end{pmatrix} \exp[ik(r\sin\theta\cos(\nu - \varphi) + z\cos\theta)] \sin\theta d\theta d\varphi,
\tag{1}
$$

where $(\rho, \nu, z)$ are the cylindrical coordinates in the focal region, $(\theta, \varphi)$ are spherical angular coordinates of the focusing system's output pupil, $\sin(\theta_{max}) = NA$ is the numerical aperture of the system, $B(\theta, \varphi)$ is the transmission function, $k = 2\pi/\lambda$ is the wavenumber, $\lambda$ is the wavelength of radiation, and $f$ is the focal length.

In the paraxial case, the polarization transformation could be observed only due to the interaction of the source beam polarization vector $\mathbf{c}(\varphi) = (\mathbf{c}_x(\varphi), \mathbf{c}_y(\varphi))^{\mathrm{T}}$ with transmission

function $B(\theta, \varphi)$. As a rule, such transformations are carried out by the use of vortex phase anisotropy of the transmission function:

$$B(\theta, \varphi) = R(\theta)\exp(im\varphi), \tag{2}$$

where $m$ is the vortex phase order, also known as topological charge.

Then, Equation (1) can be simplified as follows:

$$\mathbf{E}_m(\rho, \nu, z) = -ikf \int_0^{\theta_{max}} R(\theta)T(\theta)\mathbf{Q}_m(\rho, \nu, \theta)\sin\theta\exp(ikz\cos\theta)d\theta, \tag{3}$$

the form of $\mathbf{Q}_m(\rho, \nu, \theta)$ depends on the type of polarization, in particular, for circular polarization:

$$\mathbf{Q}_m(\rho, \nu, \theta) = \frac{i^m \exp(im\nu)}{\sqrt{2}}\begin{pmatrix} J_m(t) \\ \pm iJ_m(t) \end{pmatrix}, \tag{4}$$

where $t = k\rho\sin\theta$ [35–37] and $J_m(t)$ is the $m$th order Bessel function of the first kind.

Note, combining a vortex phase with CVBs allows for the generation of a sharper focal spot in the focal plane than a classic, radially polarized beam [38,39].

Let us consider the linear ($m = 0$) or spiral ($m \neq 0$) axicon with the transmission function:

$$B(\theta, \varphi) = \exp(ik\alpha_0 f \sin\theta)\exp(im\varphi), \tag{5}$$

where $\alpha_0$ is an axicon parameter associated with the numerical aperture of the optical element.

It is well known that axicons [40,41] are widely used for the generation of Bessel beams [42–44], which significantly change their spatial distribution in the focal domain, transforming into a narrow light ring [45–47] whose radius depends only on the angle of the inclination of conical waves and does not depend on the vortex phase component [48,49]. This property of axicons has been used for the generation of so-called "perfect" optical vortex (POV) beams [50,51], ring-shaped singular beams whose diameter does not scale with their topological charge. POV beams are used in different applications, including laser trapping and manipulation [52–55]. However, in this paper we are interested not only in the light distributions in the focal plane, but also in the area around it: in this case, it is possible to shape light distributions in the form of an optical bottle beam [56–58]. In [28], it was shown that, depending on the type of axicon used (linear, binary, or spiral), it is possible to form optical bottles that are closed on both sides or open on one side. In this case, it is necessary to change the structure of the axicon, i.e., either change the utilized DOE or use a dynamically configurable SLM. In this paper, for this purpose, we proposed to use the polarization state of the incident laser radiation and a DOE, which is a combination of different axicons:

$$B(\theta, \varphi) = \sum_{p=0}^{N} c_p \exp(ik\alpha_{0p} f \sin\theta)\exp(im_p\varphi)\exp\left(i\beta_p f \sin\theta\cos\varphi\right), \tag{6}$$

where $c_p$ are complex coefficients and $\beta_p$ corresponds to the carrier spatial frequency of $p$-th axicon in the combination. For spatial splitting or intersection of focal rings, we can just vary the value of carrier spatial frequencies $\beta_p$.

To design the binary DOE, we used the phase encoding method [59,60]. For example, to combine three axicons—one on-axis binary axicon with $m_0 = 0$ and two off-axis spiral axicons with $m_2 = -m_1 = m$—we can use a DOE with the following transmission function:

$$B_{trax}(\theta, \varphi) = \text{sgn}[\cos(k\alpha_0 f \sin\theta + m\varphi + \beta f \sin\theta\cos\varphi) + \cos(k\alpha_0 f \sin\theta)]. \tag{7}$$

All three axicons have the same parameter ($\alpha_0$), while one binary axicon is formed along the optical axis (which corresponds to the second term in Equation (7)), and two

vortex axicons with opposite vortex directions ($\pm m$) and parameters ($\pm\alpha_0$) are propagated at symmetric angles to the optical axis proportional to $\pm\beta$. Due to this symmetry, two complex-conjugate terms $\exp[\pm i(k\alpha_0 f \sin\theta + m\phi + \beta f \sin\theta \cos\phi)]$ can be combined into one term, which is the first in Equation (7).

In this case, for spatial splitting of focal rings, the value of carrier spatial frequency should be $\beta > 2k\alpha_0$.

### 2.2. Experimental Setup

The optical setup for the experimental investigation of the focusing of the laser radiation with different polarizations by the composite DOEs with a transmission function calculated by Equation (6) is shown in Figure 1A. For the fabrication of the designed DOEs, e-beam lithography was used. An example of the fabricated high-quality DOE, with a diameter of 5 mm, a microrelief height of $650 \pm 10$ nm corresponding to $\pi$-phase shift at 532 nm (the refractive index of the used resist is approximately 1.41), and a side-wall inclination angle of $3 \pm 1$ degrees is shown in Figure 1B. The presented DOE was designed using Equation (7) and has the following parameters: $m = 1$, $\alpha_0 = 0.0045$, and $\beta = 110$ mm$^{-1}$. It should be noted that the designed DOEs have a fairly simple profile with a relatively large zone size (>20 μm). Thus, for the manufacture of such elements, it is possible to use other manufacturing technologies, for example, photolithography to increase the production speed.

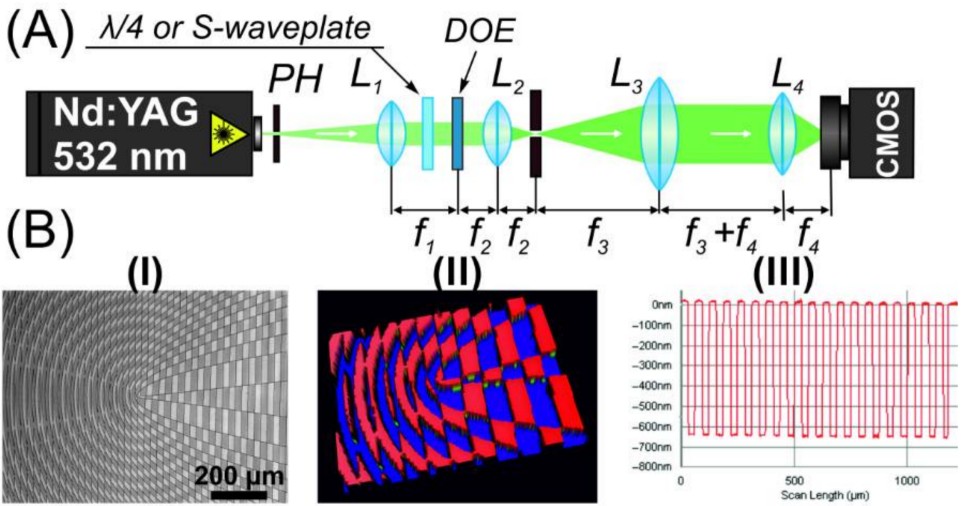

**Figure 1.** Experimental investigation of polarization transformation performed by the fabricated diffractive axicons and their combinations. (**A**) Experimental setup. (**B**) Example of the fabricated DOE: (**I**) an optical microscopy image, (**II**) an optical profilometry image of the central part of the manufactured element, and (**III**) a measured profile of the fabricated element along a vertical line.

In the experiments, the initial laser beam was extended and spatially filtered by a system composed of a pinhole (PH—aperture size of 40 μm) and a lens (L$_1$—focal length $f_1 = 150$ mm). The collimated linearly polarized laser beam with a Gaussian profile of the intensity distribution (waist diameter is approximately 3 mm) was transformed into a circularly polarized or cylindrically polarized laser beam using a quarter-wave plate or a commercially available S-waveplate (Altechna, clear aperture diameter of 4 mm). For the generation of higher order CVBs, combinations of S-waveplates and a half-wave plate were used [61]. Then, a wavefront of the formed laser beam was modulated using the fabricated DOE. A combination of two lenses, L$_2$ ($f_2 = 150$ mm) and L$_3$ ($f_3 = 200$ mm), and a diaphragm was used for spatial filtering of the shaped laser beam. Finally, the generated beam was focused by a lens L$_4$ ($f_4 = 150$ mm) and imaged by the sensor of the CMOS-video camera (ToupCam, 3328 × 2548 pixel resolution).

## 3. Results

For the design of DOEs, the following parameters were used: focal distance $f$ = 150 mm, wavelength $\lambda$ = 532 nm, axicon parameter $\alpha_0$ = 0.0015, carrier spatial frequency $\beta$ was varied in the range 30–50 mm$^{-1}$ (to ensure the formation of intersecting or non-intersecting focal rings), and the radius of the illuminating Gaussian beam was 1.5 mm.

Figure 2 shows the simulation results for various types of individual axicons illuminated by a circularly polarized Gaussian beam. As can be seen, the formation of optical bottle beams is provided by binary axicons [28]. In this case, the spiral structure of the axicon leads to the formation of a spiral structure of light rings in the focal plane. By changing the level of binary encoding [59] and fill-factor [62], one can vary the energy ratio between the formed focal ring and the central light spot (last row of Figure 2). It also allows one to change the volume and depth of the dark region of the generated optical bottle. Red circles with arrows indicate the direction of circular polarization. It can be seen that the direction of polarization does not change, so variations in the phase on focal distributions lead only to a rotation of the polarization ellipse orientation.

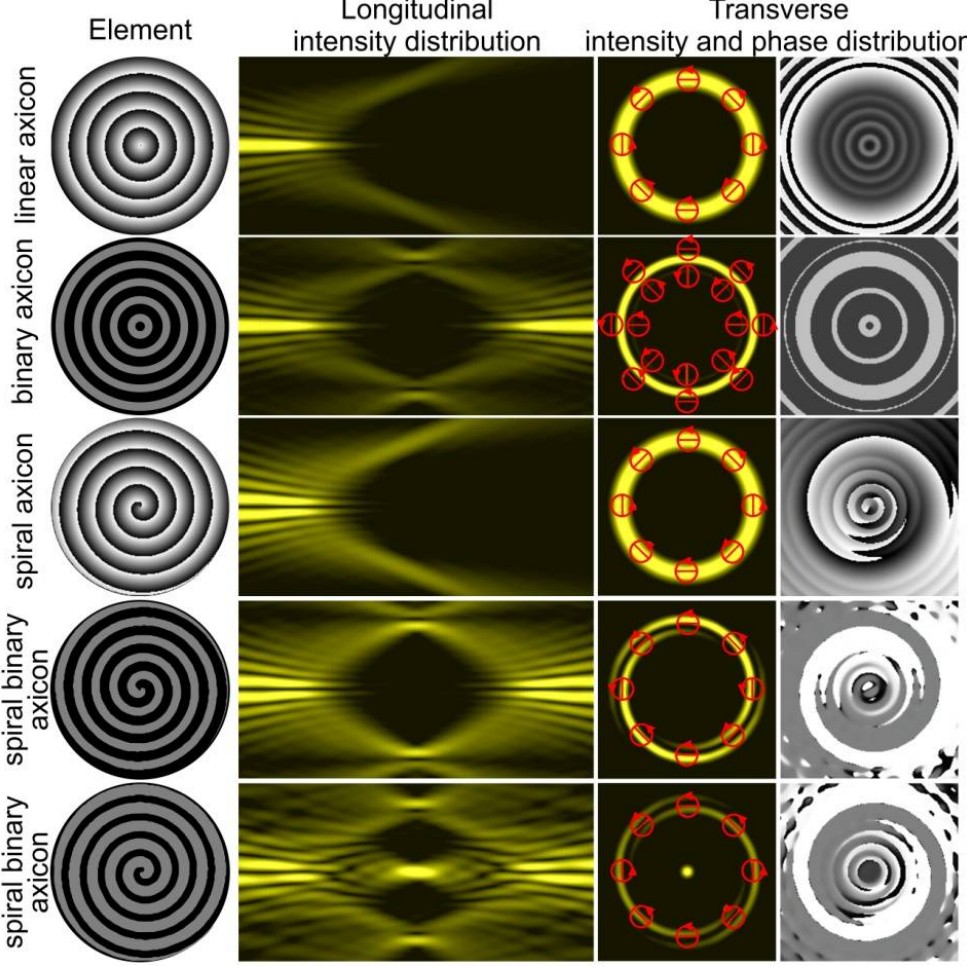

**Figure 2.** Examples of single vector optical bottle beams generated using different axicons illuminated by a circularly polarized Gaussian beam.

The modelling results for the DOE with transmission function calculated by Equation (7) with parameters $m$ = 1 for circular polarization of incident radiation are shown in Figure 3. It can be seen that the designed DOE shapes different spatial distributions before and after the focal plane. Note, similar results will be obtained for the incident radiation with another uniform polarization state (circular or linear).

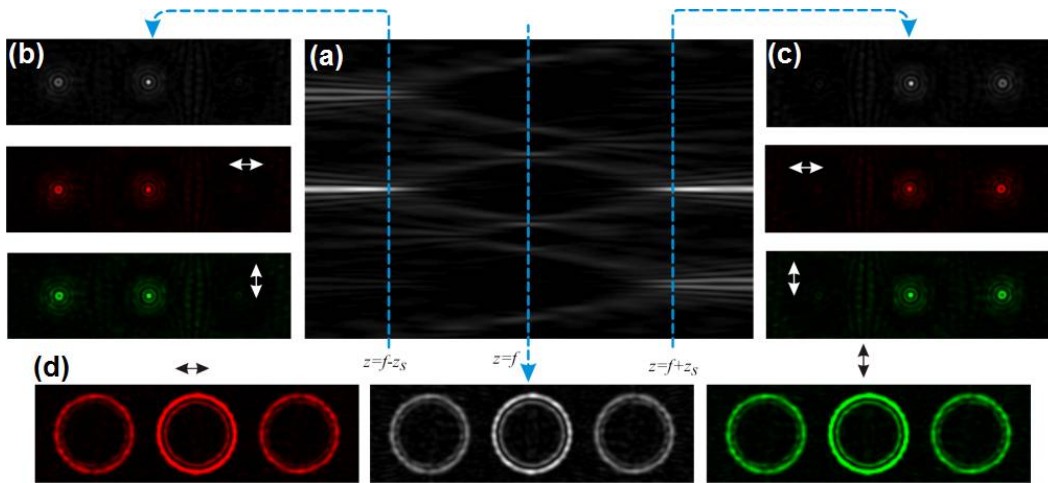

**Figure 3.** Spatial distributions for a combination of a lens and a binary DOE with transmission function calculated by Equation (7) in the case of circular «+»-polarization (red color corresponds to *x*-component, green color corresponds to *y*-component): (**a**) longitudinal field distribution in focal domain; (**b**,**c**) transversal field distribution at axis before ($z = f - z_s$) and after ($z = f + z_s$) focal plane, respectively; (**d**) transversal field distribution in the focal plane ($z = f$).

The modelling results for the case of a first-order CVB (azimuthally polarized one) are shown in Figure 4. Similar result will be obtained for the incident radiation with the radial polarization.

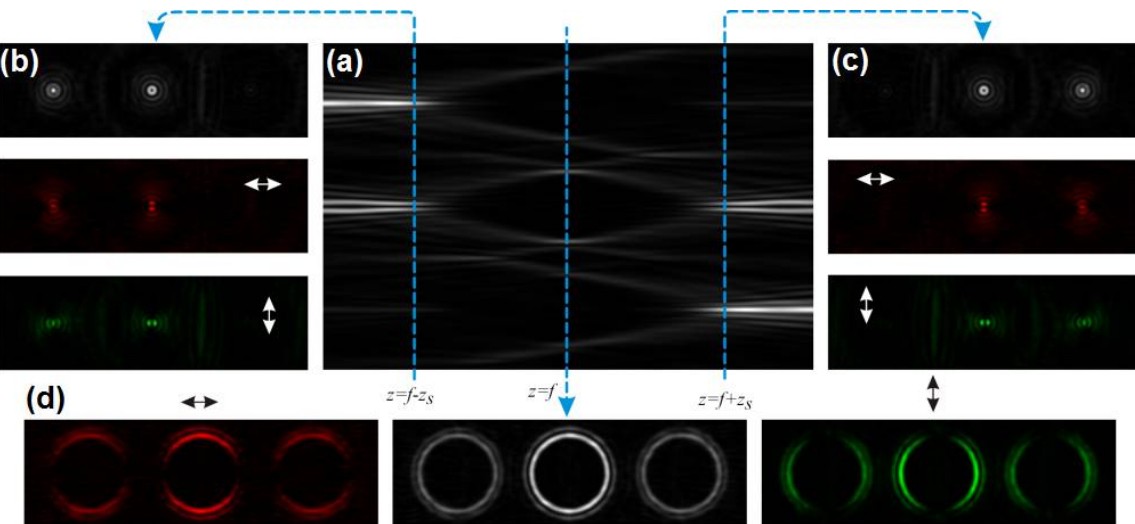

**Figure 4.** Spatial distributions for a combination of a lens and a binary DOE with transmission function calculated by Equation (7) in the case of azimuthal polarization (red color corresponds to *x*-component, green color corresponds to *y*-component): (**a**) longitudinal field distribution in focal domain; (**b**,**c**) transversal field distribution at axis before ($z = f - z_s$) and after ($z = f + z_s$) focal plane, respectively; (**d**) transversal field distribution in the focal plane ($z = f$).

A comparison of the results shown in Figures 3 and 4 shows that the light field structure in the focal plane changes slightly when the vortex phase order is changed. The focal rings' radii do not change at all, because they depend only on parameter $\alpha_0$ and focal distance *f*. This fact is well known [48,49], and is used to generate POV beams [50,51]. The polarization state affects the redistribution of energy in the double ring, but the effect of polarization can only clearly be seen with the use of an additional polarization analyzer. However, the field distributions in domains before and after the focal plane strongly depend both on the order of the vortex phase and the polarization state. In the case of circular (or linear) polarization, the three-dimensional dark region generated on the optical axis is a

completely closed optical bottle, while the off-axis generated optical "bottles" are, in fact, not completely closed due to the presence of singularity points generated by superimposed spiral binary axicons. If a vortex phase is present in the illuminating beam, the distribution before and after the focal plane will change. Moreover, by the presence (1) or absence (0) of correlation peaks in the corresponding positions (detected correlation code), one can unambiguously determine the presence of a first-order vortex singularity (and its sign) in the incident beam (see Table 1). In order to unambiguously determine vortex singularities of higher orders, the optical element should be complicated by additional vortex axicons (with corresponding vortex orders) or encoded as in [60].

**Table 1.** Results of detection of vortex phase presence (and sign) in the incident radiation with a uniform (circular or linear) polarization state.

| Incident Vortex, *l* | Intensity Distributions before and after the Focal Plane | Detected Correlation Code |
|:---:|:---:|:---:|
| *l* = 0 |  | [0110] |
| *l* = 1 |  | [0001] |
| *l* = −1 |  | [1000] |
| *l* = 2 |  | [0000] |
| *l* = −2 |  | [0000] |

Another situation is found in the case of an azimuthally polarized illuminating laser beam—the polarization-phase interaction leads to the generation of light needles before or after the focal plane for off-axis light beams. The results of the influence of the presence of the vortex phase in the illuminating beam with cylindrical polarization (radial or azimuthal) on the intensity patterns before and after the focal plane are shown in Table 2.

As can be seen from the results shown in Tables 1 and 2, intensity distributions before and after the focal plane clearly differ for different combinations of the phase and polarization states of the incident beam. Note, the coincidences in the location correlation peaks (detected correlation codes) for different phase-polarization states are accompanied by different intensity distributions, so with the complication of an element with additional vortex components (spiral axicons), it will allow a complete recognition in analogy with [62].

Thus, the designed DOEs could be used not only for the generation of complicated three-dimensional intensity distributions for optical micromanipulation, but also as a conditional detector of the polarization-phase state of incoming radiation.

Figures 5 and 6 show experimentally obtained intensity distributions for different components of the generated beams in the focal plane of the lens. A polarizer was utilized for analyzing the separated transverse electromagnetic field components. It is clear that the polarization structures of the generated laser beams are in good agreement with the results of the numerical calculations.

Figure 7 shows simulation and experimental results for multiplied optical bottle beams generated by encoded versions of combined axis-symmetric and spiral axicons illuminated by a circularly polarized Gaussian beam. As can be seen, by varying the β parameter, it is possible to form both sets of individual optical bottles and their intersections. The intersections allow the formation of an additional set of "smaller" bottles. Moreover, the volume and depth of such additional bottles can be dynamically varied by the β parameter. It can be seen that, as in the cases of single spiral axicons, the generated optical bottles are in fact not completely closed, three-dimensional dark regions. However, the diameters of

the dark regions of the generated three-dimensional regions at the ends of the bottle are significantly smaller than the diameters of the focal dark regions. Previously, such optical bottles, that were not completely closed, were successfully used for the laser trapping and guiding of airborne particles [63].

**Table 2.** Results of detection of vortex phase presence in the incident radiation with a cylindrical (radial or azimuthal) polarization state.

| Incident Vortex, *l* | Intensity Distributions before and after the Focal Plane | Detected Correlation Code |
|:---:|:---:|:---:|
| $l = 0$ | | [1001] |
| $l = 1$ | | [0110] |
| $l = -1$ | | [0110] |
| $l = 2$ | | [0001] |
| $l = -2$ | | [1000] |

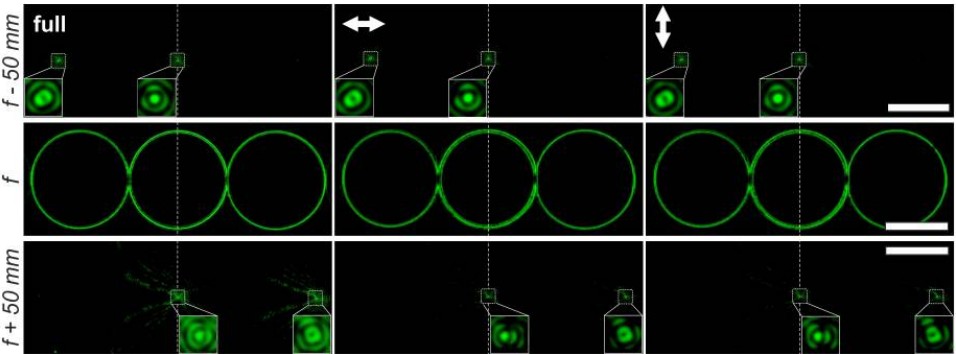

**Figure 5.** Experimentally obtained transverse component distributions of the focused circularly polarized laser beam passed through the fabricated DOE. The intensity distributions were formed in the focal plane of the lens $L_4$. The white arrow shows the position of the axis of the polarizer-analyzer. The scale bar is 1 mm.

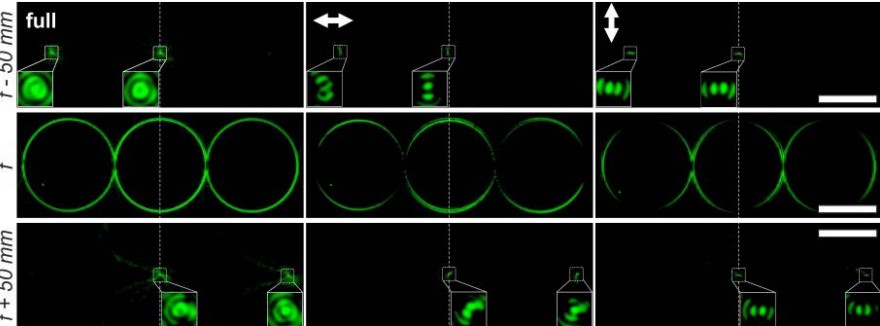

**Figure 6.** Experimentally obtained transverse component distributions of the focused azimuthally polarized laser beam passed through the fabricated DOE. The intensity distributions were formed in the focal plane of the lens $L_4$. The white arrow shows the position of the axis of the polarizer-analyzer. The scale bar is 1 mm.

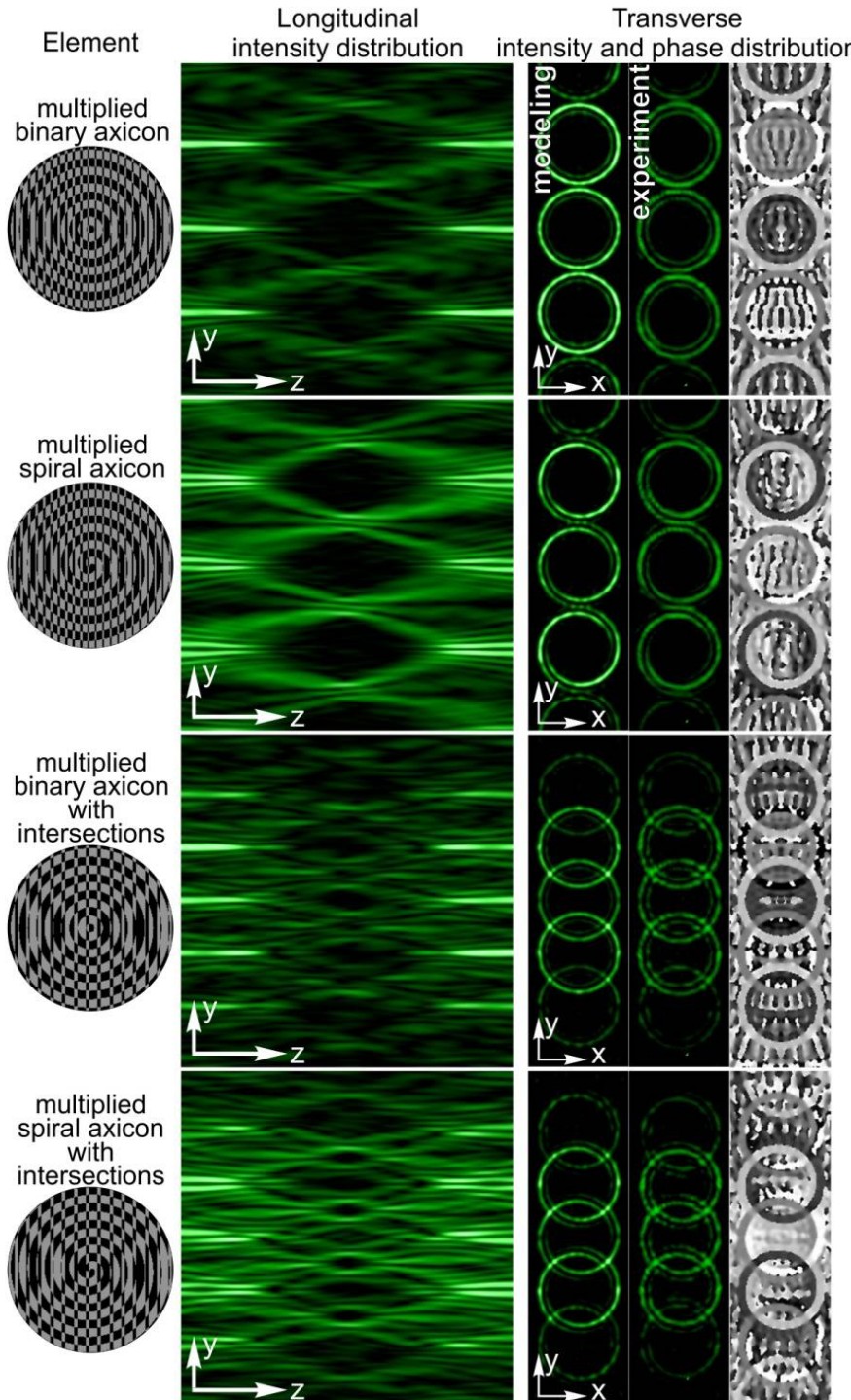

**Figure 7.** The examples of multiple vector optical bottles generated using different multiplied axicons illuminated by a circularly polarized Gaussian beam.

Changing the polarization of the illuminating beam provides additional degrees of freedom in controlling the characteristics of the optical bottle beams. Figure 8 demonstrates the influence of the type of polarization of the beam illuminating DOE, which is a combination of three axicons designed using Equation (7). It can be seen that CVBs are characterized by the different spatial distribution of different components of the vector field. In this case, using an additional polarizing filter, it is possible to dynamically remove and restore individual segments of the walls of the optical bottles, i.e., to 'open' and 'close' three-dimensional dark regions of optical bottles not only on the optical axis, but also in

the lateral regions. In addition, it is possible to use a combination of a half-waveplate and an S-waveplate for switching the polarization state of the radiation between radial and azimuthal polarization and thus affect the trapped particles.

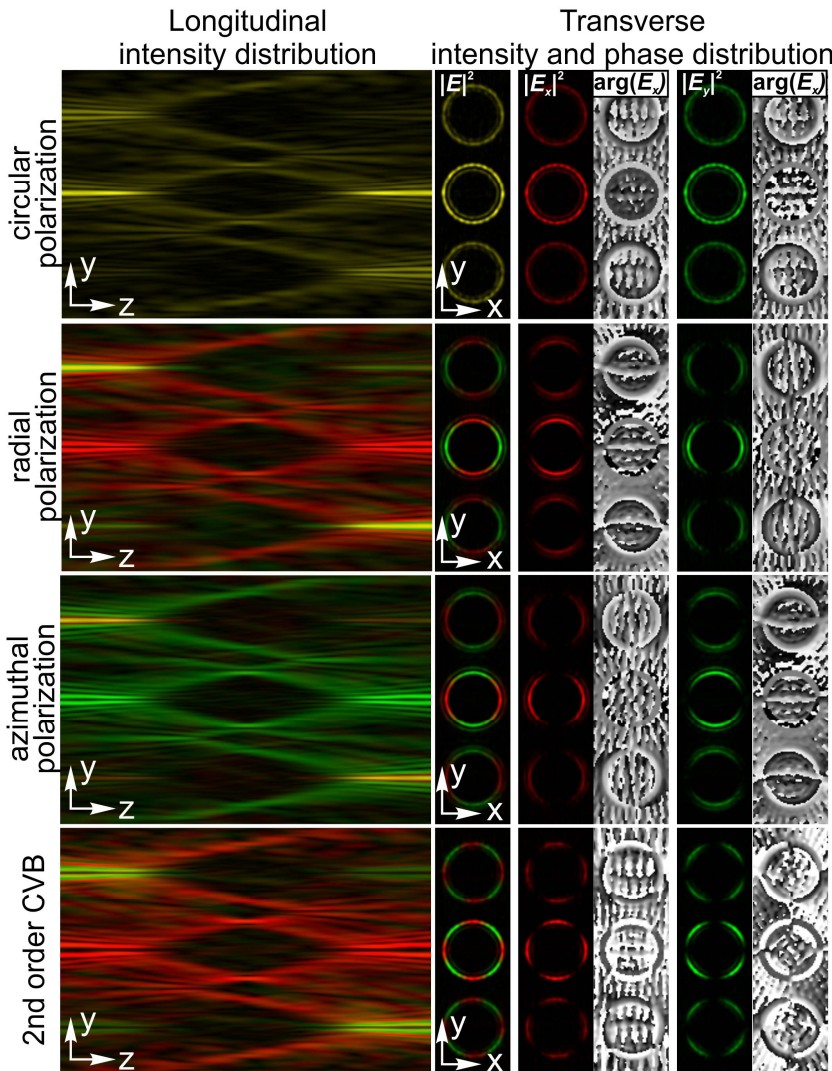

**Figure 8.** Demonstration of interaction of different polarization states with the combined binary and helical linear axicons of Equation (7). The red color indicates the x-component of the intensity and the green color indicates the y-component. The yellow color corresponds to circular polarization.

Figure 9 shows the results of modelling and experiments for two intersecting axicons of different types (axisymmetric or spiral of different orders) illuminated by CVBs of different orders. An increase in the polarization order further complicates the structure of the walls of the generated optical bottle beams, expanding the possibilities for trapping microparticles. In addition, this leads to the more complicated morphologies of the formatted LIPSS when the ultrafast laser impulses are present on the surface of different materials—it is well known that in this case the orientation of the generated structures is defined by the local orientation of the polarization state. So, using the radially or azimuthally polarized laser beams, the LIPSS are oriented in the radial or azimuthal directions. The polarization state of higher-order CVBs can be defined as $(c_x(\varphi), c_y(\varphi))^T = (\cos(p\varphi + \varphi_0), \sin(p\varphi + \varphi_0))^T$, where $p$ is the polarization order, $\varphi$ is the angular coordinate in the initial plane, and $\varphi_0$ is a constant. That means that the changes in the local orientation of the polarization in the azimuthal direction occur $p$th times more often. The possibility of using higher-order CVBs was previously demonstrated in [5], where it was shown that higher-order CVBs can lead to the

generation of surface structures with a rather complex distribution of ripples and grooves. The designed DOEs allow one to change the distance between the generated CVBs, and thus to provide an additional degree of freedom for imprinting surface structures with unconventional shapes.

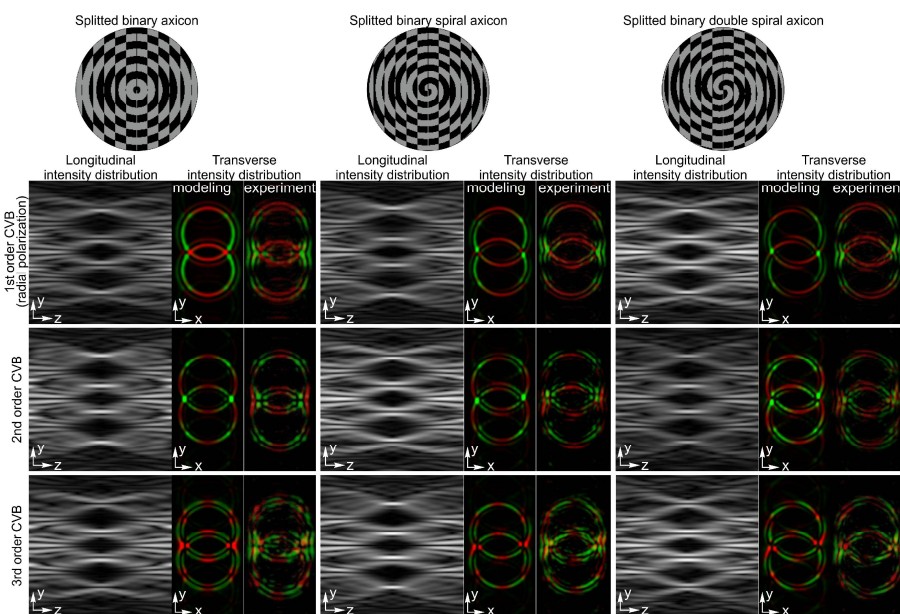

**Figure 9.** Generation of polarization controlled triple vector optical bottles using different split axicons. The red color indicates the x-component of the intensity and the green color indicates the y-component.

## 4. Discussion

The results obtained clearly demonstrate the possibility of the use of complex DOEs for splitting structured laser beams and the generation of multiple vector bottle beams, which are special types of three-dimensional optical traps with controlled distributions of intensity and polarization. However, we believe that the results will be useful not only in the field of optical trapping and manipulation when the polarization structure of the generated optical trap affects the trapping efficiency and can define the direction of the movement of the trapped particles, but also in the field of laser-matter interaction [5,6,64] for imprinting LIPSS with unconventional shapes depending on the local polarization state of the laser radiation used. In the field of laser-matter interaction, the proposed structured light fields can also be used for the fabrication of three-dimensional structures inside the volume of isotropic polymers [21] and for the investigation of complex action of intensity and polarization on the process of nano- and microstructure formation. The latter is especially important, since the morphology of the formed structure is influenced by both of these characteristics.

In this paper, we used DOEs for the realization of designed elements generating multiple vector optical bottle beams. The use of DOEs has some advantages in comparison with SLMs—this is not only their higher threshold and lack of need to use additional optimization encoding for the realization of polarization changes. The higher resolution of DOEs fabricated using e-beam lithography and photolithography techniques allow one to realize such structured laser beam generators as elements of micro-optics and use them in integrated optics devices. Now, such integrated optical devices are widely used for lab-on-a-chip biosensing applications [65]. In this case, the merging of the transmission functions of the designed axicons with the transmission functions of high-aperture lenses provides micro-optical elements, generating an array of optical traps for the investigated particles. Such microelements can also be fabricated on an optical fiber probe for selective trapping of nano- and microparticles and cells from a particle mixture or human blood

solution [66]. In addition, the higher damage threshold and efficiency of DOEs in comparison with SLMs allow one to use cheaper, low-power laser radiation sources, which makes it possible to reduce the deterioration of optical elements in the used optical setups. This is especially important when working with high-frequency pulse laser systems used in material processing applications. The high repetition rate and high laser power lead to the appearance of various defects in optical elements.

## 5. Conclusions

We numerically and experimentally investigated the spatial dynamics of the 3D intensity distribution of optical vector bottle beams depending on the polarization state of the illuminating laser radiation passed through an optical system consisting of a lens and the proposed binary DOEs combining several axicons of different types (axis-symmetrical and spiral) in the paraxial regime.

The experimentally obtained results attained with the help of the element fabricated using e-beam lithography proved the formation of complex 3D intensity distribution with the incident laser beam in the case of circular and cylindrical polarization of different orders.

**Author Contributions:** Conceptualization, S.N.K., V.S.P.; methodology, S.N.K., V.S.P., S.S., and M.D.; software, S.G.V. and S.N.K.; validation, S.N.K., A.P.P., and V.S.P.; formal analysis, S.N.K. and A.V.U.; investigation, S.N.K., S.G.V., S.A.F., and A.P.P.; resources, V.S.P., S.S., and M.D.; data curation, S.G.V.; writing—original draft preparation, S.N.K. and A.P.P.; writing—review and editing, S.N.K. and A.P.P.; supervision, S.N.K. and V.S.P.; funding acquisition, S.N.K. All authors have read and agreed to the published version of the manuscript.

**Funding:** This research was funded by the Russian Science Foundation, grant number 21-79-20075.

**Institutional Review Board Statement:** Not applicable.

**Informed Consent Statement:** Not applicable.

**Data Availability Statement:** Not applicable.

**Acknowledgments:** We would like to thank Uwe Hübner, Department of Competence, Center for Micro- and Nanotechnologies of Leibniz Institute of Photonic Technology Jena for the fabrication of the diffractive axicons.

**Conflicts of Interest:** The authors declare no conflict of interest related to this article.

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
