# Peer review of "Generation of Multiple Vector Optical Bottle Beams"

_photonics, doi:10.3390/photonics8060218_

Round 1
Reviewer 1 Report
The reviewed article can be considered the next in a series of works by the same team on light beams with a given polarization distribution and designed shape of a focus. The authors can be considered as one of the leading groups on this subject and they have published already many important results on this topic.
The present paper is a direct extension of the previous one, where light distributions formed by a tandem of a lens and an axicon with or without topological charge were analysed and experimentally realized [1]. Now authors are replacing the transmittance of a single axicon with a diffractive optical element containing their sum with added carrier spatial frequencies.
As it used to be done in similar works, the vector diffraction theory according to the Richards and Wolf approach was used in the simulation. The obtained results were then confirmed by the experimental results, for which a dedicated diffraction element containing the sum of axicons was developed and manufactured with the help of e-beam lithography.
Passing from description to opinion, I would like to state that I find the paper well written and interesting, as is usually the case with this team. In my opinion the reviewed manuscript represents an important contribution to the existing knowledge and should be of interest to readers of Photonics Journal.
On the other hand optical bottles are a hot topic that is of great interest to numerous research groups, so it is difficult to comment fairly on a non-review paper on all the results that appear, even if close and relevant. The efforts of the authors who have added an extensive list of references to the article should be appreciated, and therefore, with some embarrassment, I submit the following comments. Namely, I consider it advisable to quote the first works on spiral diffractive elements [2], [3]. Analogously, taking into account some of early works on the use of axicons to form specific focal distributions would be appropriate [4]. At last some recent results published in [5-7] seem close to the topic covered in the paper and a brief explanatory note would be also appropriate.
Concluding, the work contains new results, which supplement the outcomes hitherto published, partly in earlier papers of the authors. In my opinion the reviewed manuscript can be published in Photonics Journal provided the above remarks will be dealt with.
References:
- N. Khonina and A. P. Porfirev, "3D transformations of light fields in the focal region implemented by diffractive axicons", Appl. Phys. B 124, 191 (2018)
- Dyson, "Circular and spiral diffraction gratings," Proc. R. Soc. London Ser. A 248, 93-106 (1958)
- N. Khonina, V. V. Kotlyar, V. A. Soifer, M. V. Shinkaryev, and G. V. Uspleniev, Trochoson, Opt. Commun.91, 158-162 (1992)
- B. Rajesh, Z. Jaroszewicz, and P. M. Anbarasan, "Improvement of lens axicon’s performance for longitudinally polarized beam generation by adding a dedicated phase transmittance," Opt. Express 18, 26799-26805 (2010)
- Lei Han, Sheng Liu, Peng Li, Yi Zhang, Huachao Cheng, Xuetao Gan, and Jianlin Zhao, "Managing focal fields of vector beams with multiple polarization singularities," Appl. Opt. 55, 9049-9053 (2016)
- Zhongsheng Man, Zhidong Bai, Shuoshuo Zhang, Jinjian Li, Xiaoyu Li, Xiaolu Ge, Yuquan Zhang, and Shenggui Fu, "Focusing properties of arbitrary optical fields combining spiral phase and cylindrically symmetric state of polarization," J. Opt. Soc. Am. A 35, 1014-1020 (2018)
- Sheng Liu, Shuxia Qi, Yi Zhang, Peng Li, Dongjing Wu, Lei Han, and Jianlin Zhao, "Highly efficient generation of arbitrary vector beams with tunable polarization, phase, and amplitude," Photon. Res. 6, 228-233 (2018)
Author Response
Thank you for your favorite comments. The first reference was cited as Ref. 28. We added the rest of the mentioned references to the text.

Reviewer 2 Report
The Authors describe their achievements in generation of vector optical bottle beams with the use of DOE manufactured with electron beam lithography technique. Experimental results are compared with numerical calculations and are in good agreement with each other. The manuscript contains informative figures and is written in a concise, understandable manner. The Reviewer feels that the manuscript covers important and valuable topic since there is a growing interest in vector beams in recent years. Below there are some additional comments and questions. General questions are asked mainly out of curiosity. Technical comments point out the shortcomings noticed in text.
General questions:
- In the Introduction there is a very brief mention about SLMs and benefits of DOEs over SLMs. However, SLMs are widely used nowadays, especially in optical trapping, which was mentioned several times as the potential field of vector bottle beams application. Therefore there should be more discussion (e.g. in Section 3 or 4) on the possibilities and limitations connected with implementation of the vector optical bottle beams generation to holographic techniques. E.g. is it possible to generate a relevant hologram producing fully 3D vector bottle beam?
- Is there a way to go smoothly from one-side-opened bottle to fully closed bottle using this kind of bottle beam generation? It would be beneficial for optical trapping to first approach a particle with the open side and then enclose it inside a 3D bottle.
- The Reviewer lacks discussion on how the imperfections of the fabricated DOE affect the quality of the beam. Would less precise yet faster techniques, e.g. nanoimprint or optical lithography, be able to produce bottle beams of the comparable quality?
Technical comments:
- Figure 3: the caption reads “yellow colour corresponds to sum of (…)”. There is no yellow colour in that figure – should be white instead. Moreover, the caption contains references a), b), c), d) which are absent in the figure.
- Lines 113-114: (about parameter beta_p) “corresponds to carrying the spatial frequency of each p-th axicon in the combination”. The expression “carrying spatial frequency”, which is repeated several times in the paper, should be probably “carrier spatial frequency”.
- Figures 5-6 seem too small to be comfortably looked at. Laser spots at f=+50 and f=-50, even after magnification, are too tiny to see the details.
- Figure 1(III): it should be mentioned along which line the profile was taken
Author Response
We are thankful to reviewers for their useful comments and suggestions, which allow us to improve the quality of the manuscript making it more clear for readers. We believe the corrections made address the Reviewer’s concerns making the manuscript suitable for publication in the journal.
All changes in the manuscript are highlighted by yellow color.

Reviewer 3 Report
Please see the attachment.

Author Response
We are thankful to reviewers for their useful comments and suggestions, which allow us to improve the quality of the manuscript making it more clear for readers. We believe the corrections made address the Reviewer’s concerns making the manuscript suitable for publication in the journal.
All changes in the manuscript are highlighted by yellow color.
We have also made efforts to improve the grammar of the text.
